# Volatile-Based Diagnosis for Pathogenic Wood-Rot Fungus *Fulvifomes siamensis* by Electronic Nose (E-Nose) and Solid-Phase Microextraction/Gas Chromatography/Mass Spectrometry

**DOI:** 10.3390/s23094538

**Published:** 2023-05-06

**Authors:** Jhing Yein Tan, Ziteng Zhang, Hazirah Junin Izzah, Yok King Fong, Daryl Lee, Marek Mutwil, Yan Hong

**Affiliations:** 1School of Biological Sciences, Nanyang Technological University, 60 Nanyang Drive, Singapore 637551, Singapore; jhingyein.tan@ntu.edu.sg (J.Y.T.); zzhang043@ntu.edu.sg (Z.Z.); b200059@e.ntu.edu.sg (H.J.I.); mutwil@ntu.edu.sg (M.M.); 2National Parks Board, 1 Cluny Road, Singapore Botanic Gardens, Singapore 259569, Singapore; fongyokking168168@yahoo.com (Y.K.F.); daryl_lee@nparks.gov.sg (D.L.)

**Keywords:** volatile-based diagnosis, electronic nose (e-nose), Cyranose 320, *Fulvifomes siamensis*, SPME GC-MS, signature volatile, nitrogen for baseline, 1,2,4,5-tetrachloro-3,6-dimethoxybenzene, concentration specific clustering, fungi-host interaction

## Abstract

Wood rot fungus *Fulvifomes siamensis* infects multiple urban tree species commonly planted in Singapore. A commercial e-nose (Cyranose 320) was used to differentiate some plant and fungi volatiles. The e-nose distinctly clustered the volatiles at 0.25 ppm, and this sensitivity was further increased to 0.05 ppm with the use of nitrogen gas to purge the system and set up the baseline. Nitrogen gas baseline resulted in a higher magnitude of sensor responses and a higher number of responsive sensors. The specificity of the e-nose for *F. siamensis* was demonstrated by distinctive clustering of its pure culture, fruiting bodies collected from different tree species, and in diseased tissues infected by *F. siamensis* with a 15-min incubation time. This good specificity was supported by the unique volatile profiles revealed by SPME GC-MS analysis, which also identified the signature volatile for *F. siamensis*—1,2,4,5-tetrachloro-3,6-dimethoxybenzene. In field conditions, the e-nose successfully identified *F. siamensis* fruiting bodies on different tree species. The findings of concentration-based clustering and host-tree-specific volatile profiles for fruiting bodies provide further insights into the complexity of volatile-based diagnosis that should be taken into consideration for future studies.

## 1. Introduction

### 1.1. Wood Rot Disease in Local Urban Trees

In 1967, Singapore embarked on a “Garden City” transformation into a city with lush greenery amidst the concrete jungle of roads, buildings, and residential estates [1]. By 2017, greenery occupied over 46% of the land, with a tree canopy coverage of around 30%, bringing Singapore into the top list of cities with the highest urban tree densities in the world [2,3]. Common tree species along roadsides and in parks include *Samanea saman* (Rain Tree), *Peltophorum pterocarpum* (Yellow Flame), *Casuarina equisetifolia* (Casuarina), *Tabebuia rosea* (Trumpet tree), *Swietenia macrophylla* (Broad Leaf Mahogany), and *Syzygium* grande (Sea Apple).

The warm and humid tropical climate of Singapore creates optimal conditions for the growth of diverse plant species. However, this climate can also facilitate the proliferation of pathogenic root/wood rot fungi, which are involved in the enzymatic degradation of cell wall components such as hemicellulose, cellulose, and lignin, thus undermining the structural support and integrity of trees [4]. Wounds are the common entry point for pathogenic fungi, as wounded trees take time to repair damaged tissues [5].

The central core of the tree trunk consists of heartwood, which are dead xylem cells that provide structural support. The trunk outer layer contains sapwood which are living xylem cells responsible for transporting nutrients and water throughout the tree. Wood decay may firstly begin in the sapwood and later spread into the heartwood. Pathogenic fungal hyphae secrete oxidizing enzymes (laccase, lignin peroxidase, and manganese peroxidase) which result in sapwood discoloration and reduces the sapwood’s resistance to further secondary decay by wood decay fungi. The wood decay process results in a hollow tree because the fungal infection is compartmentalized as part of a protective response of the live sapwood, allowing for the continual formation of sound wood and bark by the vascular cambium on the exterior while the wood decay fungus decays the dead wood in the interior [6]. Hence, the early stages of wood rot disease are largely asymptomatic, whereas external signs and symptoms such as the appearance of fruiting bodies, crown dieback, and cavities may only occur at the late stages of infection [6,7,8]. Moreover, these pathogenic fungi can stay with remnant decayed wood tissues in the soil even after the removal of infected trees [9,10]. If the infected soil is left untreated, surrounding healthy trees or replacement trees may still be infected through root contact [11]. Wood decay fungi can directly infect deep opened wounds with exposed dead xylem cells on roots, tree trunks, and branches.

*Fulvifomes siamensis*, a polypore fungus, is commonly found in the soil and causes decay via infection of the root system through fungal mycelia or spores. Fruiting bodies of *F. siamensis* are commonly observed near the affected tree base and were first identified in *Xylocarpus granatum* trees at Hat Khanom-Mu Ko Thale Tai National Park, Thailand, by Sakayaroj et al. [12]. A metagenomic survey of soil and diseased samples, and barcoding of fruiting bodies collected from several species of Singapore urban trees (i.e. Rain Tree, Casuarina, Khaya, Yellow Flame, Sea Apple, Angsana, and Broad Leaf Mahogany) identified multiple strains that were genetically almost identical to the *F. siamensis* reported by Sakayroj in the ITS1 barcoding region [13]. Further in vitro wood decay studies have supported its broad-spectrum pathogenicity to multiple urban trees in Singapore. Hong et al. [13] also reported that the *F. siamensis* fruiting bodies exhibited a high level of morphology plasticity at various developmental stages on different host trees: *F. siamensis* fruiting bodies of the same or nearly identical genotypes varied in color (from orange, brown, to black), the absence/presence of white fringes, and in terms of shape (clumps, multilayer, and flat fan-like structures). Such morphological plasticity of *F. siamensis* could pose a challenge to the identification and diagnosis based on morphology and visual assessments.

Internationally adopted early diagnostic assessments involve the molecular-based analysis of wood tissues and soil samples via DNA extraction and polymerase chain reactions (PCR) for specific and sensitive results [14]. However, these methods are often time-consuming, costly, and with limited coverage since each PCR consists of one set of taxon-specific primers [15]; they are also invasive for sampling wood tissues. It would also require specialized knowledge and a specific laboratory set up to carry out PCR-based diagnosis. Therefore, there is the need for cost effective and non-invasive alternative diagnostic solutions for pathogenic fungi that infect trees.

### 1.2. VOC Detection with E-Nose and SPME GC-MS

Living plants release mixtures of volatile organic compounds (VOCs) as a response mechanism with respect to the plant’s physiological health status, age, infection by microbial pathogens [16], and other biotic and abiotic interactions with the environment (i.e., light, temperature, humidity) [17,18]. These VOCs can be generally grouped into isoprenoids, benzenoids, fatty acid derivatives, and amino acid derivatives [16,19]. Fungi also emit a range of VOCs during primary and secondary metabolism, such as aldehydes, alkenes, alcohols, ketones, benzenoids, carboxylic acids, and isoprenoids. Fungal-specific VOCs can be detected and used for identification [20].

VOC-based diagnosis provides a great advantage over the traditional molecular methods, as VOC sampling can be utilized to discriminate specific fungi-infected samples. The analysis is non-destructive and does not compromise the value of the sample materials. Machine learning algorithms can also potentially be trained to have a higher discrimination power between different classes of pathogen-infected samples [21].

#### 1.2.1. SPME GC-MS

Solid phase microfiber extraction (SPME) is a solvent-free sampling technique that allows non-exhaustive volatile extraction [22], and only requires a small amount of analyte in comparison to traditional solvent-based extractions [23]. Microfiber releases analytes for gas chromatography mass spectrometry (GC-MS) analysis that identifies and quantifies each volatile [24], displaying remarkable potential in plant diagnostic applications. VOC-based detections in plant material using SPME GC-MS have been demonstrated on *Phytophthora infestans* and *Fusarium coeruleum*-inoculated potato tubers [25] and five postharvest pathogens-inoculated onion bulbs [26]. Despite the accurate quantitative and qualitative analysis, GC-MS analysis is not always readily available and affordable [27]; hence, there is the need for a simpler VOC analysis instrument such as the electronic nose (e-nose) that is more affordable and can be used in the field.

#### 1.2.2. E-Nose System

The key operating principle of the e-nose resides in the change in the resistance and electrical conductivity of its sensors in response to interactions of the sensor material with the gas-phase analytes, which belong to diverse chemical classes or have different functional groups. A panel of sensors with variable specificity for different chemicals would generate a unique response profile that can differentiate different volatile samples. The e-nose was found to be useful in discriminating *Fusarium verticillioides* inoculated in maize from healthy maize [28]. There have been few reports on the use of e-nose to detect wood rot diseases in urban tree species. Baietto et al. [29] investigated the use of a metal oxide semiconductor commercial e-nose (PEN3), which successfully discriminated between the in vitro healthy and decayed root segments of five hardwood and conifer trees species, *Aesculus hippocastanum*, *Cedrus deodara*, *Platanus* × *acerifolia*, *Quercus robur*, and *Liquidambar styraciflua*, after inoculation with three different pathogenic root rot fungi *Armillaria mellea*, *Ganoderma lucidum*, and *Heterobasidion annosum* after a 12-month incubation. 

Another study used a handheld commercial e-nose, the Cyranose 320, to demonstrate the discrimination of healthy and *Ganoderma boninense*-infected oil palm trunk and soil samples [30]. While these reports demonstrated the qualitative differentiation ability of the e-nose between diseased and healthy samples, validation by another complementary technology has often been missing. Moreover, the volatile molecules responsible for the differential detection were unknown, since the e-nose is only suitable for pattern recognition; it does not provide the specific identity and quantity of VOCs in the sample. For field applications, there is also the concern that different operational environments (such as air contaminants from automobiles and industrial effluents [31]) might introduce volatiles that mask those pathogenesis-related volatiles.

In addition to the plant disease diagnosis conducted by Markom et al. [30], the Cyranose 320 (Smiths Detection, Inc., Pasadena, CA, USA) has also been studied in terms of its usage in the fields of medical diagnosis [32], as well as food and industrial applications [33]. The Cyranose 320 sensing component is equipped with the NoseChipTM with an array of 32 polymer composite chemi-resistor sensors that are coated with conductive films arranged across electrodes [34]. The voltage change (∆R/R_o_) over every sensor in the array is measured and transduced into a resistance reading that is successively analyzed using an onboard unsupervised machine learning algorithm, Canonical Discriminant Analysis (CDA).

The Cyranose 320 has a “Purge Inlet” path that draws in a baseline gas (ambient air is often used for convenience) to the sensors for the measurement of R_o_, which is independent from the “Sample Inlet” path and which draws in the VOCs from the analyte for the measurement of R_max_. The conductive pathways will be restored back to their original state once the analyte is taken away [34]. Since VOCs are classified based on the ∆R/R_o_ data measured across the Cyranose 320’s sensor array, it is important to establish a consistent baseline gas (R_o_). Whilst the use of ambient air is the most convenient option, it would not provide consistent e-nose reading results as the ambient air composition will change based on the surrounding environment. For example, the composition of the air in the environment alongside a road would differ from that in a laboratory, inside a forest, or along an urban park. Hence, it would be unsuitable to rely on ambient air for baseline R_o_ measurement and nitrogen gas would be a good alternative. Nitrogen gas is widely used in industrial applications as a zero gas or purging gas [35]. The main reasons for this are that nitrogen gas is the main component in ambient air, it is readily available, colorless, odorless, non-flammable, non-toxic, and non-reactive (inert).

The Cyranose 320 also contains on-board diagnosis via the “Identify” and “Train” functions. The “Train” function is first used to build up a training model with five to ten readings per sample class on six known samples. After cross validation of the training model, the “Identify” function is then utilized to identify an unknown sample based on the training model. Upon successful matching of the unknown sample to one of the sample classes in the training model, the identification result can be immediately viewed on the e-nose screen with the training model’s class name together with an identification quality rating ranging from the highest five-star confidence matching quality (*****), to a three star (***), or to the lowest one star matching quality (*). In the case where the unknown sample cannot be matched to any of the sample classes in the training model, “Unknown” would be displayed on the screen [34].

#### 1.2.3. Sampling Algorithm: Canonical Discriminant Analysis (CDA)

CDA is a useful multivariate unsupervised machine learning method that can separate samples into classes in a lower dimensional discriminant space with respect to the variance of multiple measured independent variables [36]. During cross validation, the 32 sensor variables of ΔR/R_o_ data are calibrated after an input of an optimum number of Principal Components (PC) which will condense ΔR/R_o_ data from the sensor factors that capture the most variance within the data set. Data processing using the Normalization 1 tool with auto-scaling could reduce the effect of volatile concentrations and sampling technique differences on the sample volatile profile [34].

The Interclass Mahalanobis distance (MDist) is a quantitative representation of the correlation of data points from two sets of sample classes. A MDist score larger than 5.000 indicates that the two sample classes are distinct and dissimilar from each other, while a score less than 5.000 indicate that the two sample classes are indistinguishable from each other [34]. The software Chemometric Data Analysis Program (CDAnalysis) version 11.2 can be utilized to create the Canonical graph plots of the Cyranose 320-generated sensor resistance data set. Selected data sets can have their individual sensor data variance calculated to identify “active sensors” that contribute to the Canonical graph PC axes. The variance would tally up to at least 90.0%. The average of the selected active sensors response data ΔR/R_o_ is then plotted against the sensors with standard deviation as error bars to provide further insight to the resistance pattern difference among samples on the active sensors.

## 2. Objectives

With the aim to develop a non-invasive diagnosis method for the white rot polypore wood rot fungus *F. siamensis* that threaten many urban trees species in the tropics, two complementary approaches were evaluated—e-nose and SPME GC-MS. Firstly, a commercial e-nose was tested for its sensitivity and specificity for the detection of some plant- and fungi-relevant volatiles. This work also explored the use of inert nitrogen gas as a baseline gas and identified the possible contribution factors to the much-improved responses by the e-nose. The e-nose/nitrogen set up was further used to evaluate its capability in differentiating between pure cultures of *F. siamensis*, its fruiting bodies, and disease tissues collected from multiple urban tree species from reference samples, all within a 15-min sample incubation time that is more relevant to field applications. The same samples were concurrently tested with SPME GC-MS to characterize the composition volatiles in each sample that potentially underline the differential detection by e-nose. There was also an effort to identify the fungus signature volatiles and compounds consistently present in significant quantities across all *F. siamensis* samples. Finally, we created a model with a trained dataset to conduct some in-field tests to validate the utility of the e-nose in identifying the *F. siamensis* fungal fruiting bodies in various host tree species. 

## 3. Methodology

### 3.1. Field Sample Collection, DNA Isolation, Metagenomic Survey, and Sample Barcoding

Sample collection was conducted by following the protocol detailed in a previous report [13]. Briefly, fruiting bodies (FB) were dislodged from the tree with full records (i.e., the health status of the host tree, its location, time of collection, and photos of the fruiting body). Diseased tissues (DT) were also collected either from the decayed wood behind the fruiting bodies or from decayed tissues inside the trunk after tree removal. One healthy tissue (HT) was also obtained from a Casuarina tree. For each test sample, five grams were set aside for DNA isolation before molecular identification either through direct barcoding (for fruiting bodies) or metagenomic analysis of the ITS1 region (for diseased tissues) by following the same procedures previously reported [13]. Another two grams were used for SPME GC-MS analysis, while around forty grams (unless otherwise specified) were repacked into a new Glad® Freezer Gallon Zipper Bag (Glad Products, Oakland, CA, USA) for VOC tests by the Cyranose 320 e-nose. The samples are detailed in Table 1.

### 3.2. Culture for Pure Fungal Isolates

The collected *F. siamensis* and *R. microporus* fruiting body samples were dissected with a sterile handsaw and the interior was processed into small fragments for culture preparation. The processed sample fragments were inoculated onto Potato Dextrose agar (PDA) culture media with antibiotics (Streptomycin 30 mg/L + Ampicillin 100 mg/L). Subcultured plates were made through the transfer of hyphae at the colony edge onto a new PDA plate with sterile scalpel blades to obtain a pure culture. All plates were incubated at 30 °C in the dark for fungal growth. DNA isolation and PCR amplification were then conducted as per the method used in Hong et al. [13] for fungal culture barcoding. One pure isolate of *F. siamensis* (GenBank no. OQ618213) and one of *R. microporus* (GenBank no. OQ558869) were selected for e-nose and SPME GC-MS analysis, since they are among the most prevalent wood-rot pathogens in Singapore urban tree species. *R. microporus*, a white rot fungus, secretes plant cellulose biomineralization and lignin degradation enzymes, which can be identified by visible white mycelia strands that firmly adhere to the bark of tree roots [37].

### 3.3. Cyranose 320 E-Nose Method Settings

The method settings were adapted from the Sensigent practical guide [38] and are detailed in Table 2. The Cyranose 320 was pre-warmed for 5 min at the beginning of each run session at 42 °C. In this paper, refillable 3L multi-layer gas sample bags (Jensen Inert Products, Coral Springs, FL, USA) were filled with pure nitrogen gas at a purity of ≥99.99% and attached to the e-nose purge inlet valve to achieve an airtight seal during the ‘baseline purge’. A sample was incubated inside a Glad® Zipper Bag for 15 min (or another specified time) before the 1.2 mm × 40 mm snout sampling needle of the Cyranose 320 was inserted into the bag and was stopped at 2 cm away from the sample material. After measurement, raw data were downloaded and CDA Canonical graph plotting was conducted with CDAnalysis software version 11.2 utilizing 10 Principal Components (PC) for the generation of the graph.

### 3.4. E-Nose Tests

#### 3.4.1. Wood Rot Fungi and Plant-Related Volatiles: Turpentine, Farnesene, and Acetic Acid

This project started from the assessment of the Cyranose 320’s sensitivity and specificity in distinguishing Turpentine (TUR), Farnesene (FAR), Acetic Acid (AA), and air from the biosafety cabinet (AIR) volatiles using nitrogen and ambient air as baseline gases. In the biosafety cabinet, TUR, FAR, and AA liquids were introduced onto a piece of 2 × 2 × 0.04 cm L-fold paper towel (PT Suparma, Tbk, Surabaya, Indonesia) to achieve different volatile concentrations from 0.05 ppm to 5.0 ppm in 2 L of head space in a Glad® Freezer Gallon Zipper Bag with the assumption of total evaporation. The sample bags were inflated to contain about 2 L of air, sealed, and left to sit for 15 minutes prior to the e-nose analysis to allow VOC evaporation in the headspace to reach the intended concentrations.

#### 3.4.2. Fungal Mycelia Volatiles

The pure fungal isolates of *F. siamensis* (F) and *R. microporus* (R) were allowed to grow on PDA medium until the whole culture plates (90 mm diameter) were covered. The 8-mm diameter end of a 1000-μL micropipette tip was used to cut into the pure culture to create one cut plug containing approximately 3.67 mg of fresh mycelia. Cut plugs of varying quantity (denoted as −5, −10, −15, for five, ten, and fifteen cut plugs, respectively) were transferred into a Glad® Quartz-sized Zipper Bag (Glad Products, Oakland, CA, USA), which was then inflated to have 500 mL of head space, sealed, and incubated for 15 min at room temperature before e-nose analysis. Each sample was analyzed 6 times. 

#### 3.4.3. E-Nose Field Identification of *F. siamensis* Fruiting Bodies

The e-nose was used to measure and analyze four *F. siamensis* fruiting bodies (YF156FB1, YF157FB, R159FB, YF160FB), one *G. australe* fruiting body (CE153FB), and one Casuarina healthy wood tissue (CE151HT) sample. Each measurement contained 5–6 readings of the six samples, nitrogen gas was used for the baseline, and the samples had 15 minutes of incubation time. The six samples were utilized in the training model, Model A. All the samples included in the training model had been molecularly identified (Table 1). After cross validation of the training Model A, this model was then used for the field diagnosis test through the “Identify” function of Cyranose 320 in Section 4.3. A Canonical algorithm with medium identification quality, auto-scaling, and the Normalization 1 tool was used during every sample identification.

### 3.5. SPME GC-MS Procedures

The fruiting body and diseased tissue samples were cut into fine pieces using a sterile handsaw, and two grams of the sample were placed into a 20 mL clear vial capped with 20 mm Viton® Septa Seals (Supelco Inc., Bellefonte, PA, USA). For fungal culture samples, six 8-mm diameter cut plugs were similarly placed into the clear vial and capped. 

The solvent-free sampling of analytes was conducted with a SPME fiber assembly consisting of a 50/30 μm Divinylbenzene/PDMS/Carboxen-coated fiber (Supelco Inc., Bellefonte, PA, USA) inside a protective needle, which was attached to a manual SPME holder. The fiber assembly was introduced into the vial through a hole in the vial cap septum and the fiber was released from the needle to stay 1–2 cm above the sample for volatile sampling. Sampling was normally carried out in darkness at room temperature for 1 day. The fiber was retracted into the needle to stop sampling.

For GC analysis, the SPME needle was introduced by the SPME holder through a SPME microseal fitted with a molded Thermogreen LB-2 septa with an injection hole into the SPME inlet liner inside the sampling chamber before releasing the fiber, where a temperature of 250 °C allowed the compounds to be desorbed. The injection was carried out in a splitless manner. An Agilent 7890A gas chromatograph including a 30 m × 0.25 mm × 0.25 µM HP-5MS semipolar capillary column connected to an Agilent 5975C mass spectrometer (Agilent Technologies, Santa Clara, CA, USA) was used for GC-MS analysis, which lasted 50 min overall. From an initial temperature of 40 °C, the oven was heated at a rate of 6 °C/min up to 80 °C, then increased at a rate of 3.4 °C/min to reach 170 °C, before increasing at 12 °C/min to reach a peak of 300 °C, at which it was kept for 4 min. The electron ionization potential was set at 70 eV and the electron ionization (EI) source worked at a temperature of 230 °C. The temperature of the quadrupole analyzer was 150 °C. Helium circulating through the column at a steady flow of 1.2 mL/min served as a carrying gas. The MS detector was working at a mass range of 40–500 amu in positive polarity mode. Recorded electron ionization mass spectra were compared against the NIST Mass Spectral Database for analyte identification.

## 4. Results

### 4.1. Sensitivity and Specificity of E-Nose with Nitrogen Gas for Baseline

#### 4.1.1. Multiple VOC Differentiation by E-Nose

This project started with the assessment of e-nose sensitivity and specificity in differentiating three pure volatiles released either by plant or fungi (Figure 1): Acetic Acid (AA), Turpentine (TUR), and Farnesene (FAR). AA (CH_3_COOH) is a monocarboxylic acid that has been reported to be an intermediary product formed during cellulose degradation by wood-destroying fungi [39], and which was also confirmed by SPME GC-MS to be emitted by *F. siamensis* fruiting bodies (i.e. SG145FB). TUR (C_10_H_16_) is a mixture of unsaturated hydrocarbon terpenes, mostly alpha-pinene and beta-pinene, and is commonly derived from conifers. FAR (C_15_H_24_) is a mixture of isomeric sesquiterpenes, of which one isomer, (E)-beta-farnesene, has been identified to be a volatile released from pathogenic fungi *F. siamensis* through SPME GC-MS analysis. This study represents the initial examination of the electronic nose’s capability to identify pure volatile compounds associated with wood-decaying fungi. 

To assess the Cyranose 320 e-nose’s ability to distinguish the selected VOCs, canonical analysis was conducted to check the qualitative discrimination of VOCs in clustering data points for each sample into distinctive groups. The Interclass Mahalanobis distance (MDist) was also generated for the same samples for a quantitative assessment. A MDist score larger than 5.000 indicates that the two sample classes are distinct and dissimilar from each other, while a score less than 5.000 indicates that the two sample classes are indistinguishable from each other. Using ambient air for the e-nose baseline purge, the CDA results showed that all pure VOCs could be qualitatively classified and differentiated from the ‘AIR’ at sample concentrations of 5.0 ppm, 2.5 ppm, 0.5 ppm, and 0.25 ppm. There was also good specificity with clear distinctions among the three volatiles TUR, FAR, and AA, with an MDist greater than 5.000 (Figure 2A for 0.25 ppm). For VOCs of 0.05 ppm, only AA could still be distinctly clustered, with an MDist higher than 5.000 against other VOCs. The other three VOCs could not be distinctly clustered with an MDist less than 5.000 against each other. The TUR-FAR pairwise comparison had the lowest MDist value likely due to their similar molecular weight and chemical structure (Figure 2C).

With the same parameters for measurement, a pouch was used to supply pure nitrogen gas for the baseline. The same VOCs at 0.25 ppm were distinctively clustered with higher MDist values, suggesting better specificity (Figure 2B). For VOCs at 0.05 ppm, the e-nose with nitrogen for the baseline gas was able to clearly distinguish AIR, TUR, FAR, and AA volatiles apart from each other with an MDist greater than 5.000 for each pairwise comparison (Figure 2D). For TUR-FAR pairwise comparison, the MDist value was 9.967, a significant improvement over the use of ambient air for the baseline gas (2.649).

Through the statistical analysis of the raw response ΔR/R_o_ data for the 32 sensors with nitrogen for the baseline gas, sensor 6, 23, 24, 31, and 32 were identified as the main active sensors that account for the largest variance (8.151%, 6.894%, 6.881%, 37.407%, and 36.937%, respectively) across the sample readings with nitrogen for the baseline gas (Figure 3B). Sensors 23 and 24 had some negative responses (an increase in resistance upon contact with FAR, TUR, and AIR), which should increase the resolution power of the e-nose. In comparison, for the use of ambient air for the baseline gas, only sensors 31 and 32 (48.608% and 47.957%, respectively) were the active sensors, accounting for most of the variance in sample readings, while sensors 6, 23, and 24 only made negligible contributions (Figure 3A). This implies that the use of nitrogen for baseline gas could activate more sensors to respond. It was also noted that, for the same responsive sensors (31 and 32) for both ambient air and nitrogen baseline gases, the magnitude of response was significantly increased multiple times. Such an increase in the magnitude of response could be the reason for better sensitivity.

#### 4.1.2. *F. siamensis* Culture, Fruiting Body, and Diseased Tissue Differentiation by E-Nose

The e-nose with nitrogen for the baseline gas was further utilized to differentiate pure culture of *F. siamensis* isolate from that of *R. microporus* and from PDA media with no fungus growth (Figure 4A,B). The e-nose could distinctively classify fifteen *F. siamensis* fungal mycelia cut plugs (F-15) within 15 min of sample incubation (Figure 4A). There was distinctive clustering of F-15 from fifteen cut plugs of PDA agar (AGAR-15) and fifteen *R. microporus* fungal mycelia cut plugs (R-15). The high MDist value (62.636) for the R-15/F-15 pairwise comparison gives strong confidence that *F. siamensis* fungal mycelia can be clearly differentiated by the e-nose within a short sample incubation time.

Then, two *F. siamensis* fruiting bodies, one *G. australe* fruiting body, one *R. microporus* fruiting body, Casuarina diseased tissue dominated by *F. siamensis*, and one healthy wood sample (CE151HT) were also tested with the e-nose. The fruiting bodies were barcoded and the fungal composition in diseased tissues was assessed by metagenomic analysis (Table 1). Each sample was transferred into a new Glad® Freezer Gallon Zipper Bag, incubated for a selected time period before e-nose analysis with nitrogen for the baseline. The first analysis was conducted after 15 min of incubation (Figure 4C). To assess the necessity for a longer incubation time, the same samples were re-bagged and sealed after the first analysis for an extended incubation period of five days before e-nose re-analysis. With 15 min of incubation, the healthy casuarina wood was clearly differentiated from all the other samples with MDst values higher than 17. The *G. australe* fruiting body was also well separated from all other samples with MDist values higher than 17, and the same was determined for the *R. microporus* sample. For the *F. siamensis* sample, CE152DT and CE151FB were non-distinguishable from each other, which was expected because one was a diseased Casuarina wood tissue dominated by *F. siamensis* and the other was a *F. siamensis* fruiting body collected from a neighboring Casuarina tree. These results prove the good differentiation power of the e-nose for *F. siamensis* fruiting body and infected diseased tissues against the other two wood rot fungi in Singapore. It was also noted that a *F. siamensis* fruiting body (CE149FB) was distinctively clustered from all other samples, suggesting its unique volatile profile.

The same samples after five days of incubation gave almost the same classification results (Figure 4D), suggesting that a longer incubation period is not necessary for accurate diagnosis.

### 4.2. Volatile Profiles of Samples Identified by SPME GC-MS

SPME GC-MS identified the chemical structure of each VOC in the culture, fruiting body, and diseased tissue samples. The relative abundance for each volatile can also be derived by dividing the peak area of the volatile by the total area of all the detected peaks. The relative abundance of each peak is represented by the number of stars. Both the name and the unique identification number by the Chemical Abstracts Service (CAS) are used to represent each volatile. The results are summarized in Table 3. The most significant finding is the consistent presence of a major volatile, 1,2,4,5-tetrachloro-3,6-dimethoxybenzene, in all of the *F. siamensis* samples, but not in samples of *G. australe* or *R. microporus*. This can be regarded as the signature volatile for *F. siamensis*, and it may be the main contributor to the specific differentiation by the e-nose. Between the pure culture cut plugs for *F. siamensis* and *R. microporus*, most volatiles are specific for either strain. Such a difference in volatile profiles is thus consistent with the ease of differentiation by the e-nose. The *F. siamensis* fruiting body samples, CE149FB and CE151FB, did not share any significant volatiles with the *G. australe* fruiting body (CE153FB) or *R. microporus* fruiting body (P175FB3). Their distinctive profiles are consistent and might underline the distinctive clustering by the e-nose for the three fruiting bodies.

Interactions between the fungi and the host tree and growth media may have influenced the volatile profile of the samples. *F. siamensis* grown on PDA culture media released high amounts of (E)-beta-farnesene and 2,7-dichloro-1-methoxydibenzofuran, which was not detected in *F. siamensis* fruiting body or diseased tissue samples. Similarly, *F. siamensis* fruiting bodies collected from different host trees also emitted unique volatiles that are not shared by all fruiting bodies. For instance, *F. siamensis* fruiting body collected from a Yellow Flame tree uniquely released 1,2,4-trichloro-5-nitrobenzene. Volatiles contributed by the tree tissue itself could also influence volatile profiles. Indeed, 2,4-bis(chloranyl)-1,5-dimethoxy-3-methylbenzene from the diseased tissues infected by *F. siamensis* (CE149DT2, DT5) was possibly contributed by the Casuarina tree, since it was absent in fruiting bodies and pure cultures.

Unexpectedly, two fruiting bodies of *F. siamensis* collected from two neighboring Casuarina trees (CE149FB and CE151FB) did not cluster together in the e-nose canonical analysis results (Figure 4C,D). GC-MS identified some additional volatiles, such as 3,5-bis(chloranyl)-2,4-dimethoxy-6-methyl-phenol and 1,4-dichloro-2,5-dimethoxybenzene in CE149FB. It is suggested that these sample-specific volatiles could be the reason for its distinctive clustering, while the shared *F. siamensis* volatiles underline its distinctive clustering from all non-*F. siamensis* samples. The variation in volatile profiles among fruiting bodies of the same genotype could be due to developmental stages or other environmental factors.

### 4.3. Field Identification of F. siamensis Fruiting Bodies with the E-Nose

The e-nose pre-loaded with Model A was evaluated for field applications in the diagnosis of *F. siamensis* fruiting bodies. Fruiting bodies suspected to be *F. siamensis* were found in one *Samanea saman* tree and two *Peltophorum pterocarpum* trees. From each host tree, one big piece (A) and one small piece (B) of the suspected fruiting bodies were dislodged to assess the possible influence of sample size on the e-nose diagnosis. The samples were placed in a Glad® Freezer Gallon Zipper Bag and incubated for 15 min before the e-nose testing with nitrogen for the baseline gas via the “Identify” tool (Figure 5A) with the method described in Section 3.4.3. Their molecular identities were later confirmed by barcoding (Table 4). The air near the *Peltophorum pterocarpum* tree (YF2) was tested by pointing the e-nose needle near the tree stump with no surrounding fruiting bodies. This was used as a negative reference point.

The on-board e-nose “Identify” tool (Figure 5A,B) was able to accurately identify and classify all of the tested samples to a *F. siamensis* fruiting body training sample class, YF156FB1, but with varying confidence levels (Table 4).

### 4.4. Sensitivity of E-Nose to Changes in VOC Concentration

Through the CDA algorithm clustering and with nitrogen as the baseline gas, the testing of single volatiles of different concentrations (0.05, 0.25, 0.5, 2.5, and 5 ppm) revealed quantitative-based clustering, although the clustering resolution varied for different VOCs. The clustering of varying concentrations of AA (Figure 6A) and FAR resulted in five distinct classes for the five concentrations, while TUR samples were separated into four classes (Figure 6B), with samples of 0.05 ppm (TUR0.05n) and 0.25 ppm (TUR0.25n) clustered together. Similarly, a quantitative change in fungus culture (five, ten, and fifteen cut plugs) of *F. siamensis* (Figure 6C) and *R. microporus*, (Figure 6D) also led to distinctive clustering.

## 5. Discussion and Conclusions

Assessing root and wood tissue decay in trees remains a challenge. Existing diagnostic solutions vary in their invasiveness, reliability, ease of use, and cost. This paper has developed volatile-based methods of detecting a selected pathogenic fungus that threatens multiple urban trees in Singapore. A commercial e-nose, Cyranose 320, was evaluated for its sensitivity and specificity for plant- and fungi-related pure volatiles, then on VOCs emitted by *F. siamensis* pure cultures, fruiting bodies collected from multiple host tree species, as well as diseased tissues dominated by this fungus. SPME GC-MS was also used to analyze the volatile profile of the VOCs emitted by the fungal culture mycelia, fruiting bodies, and diseased tissues samples. Lastly, the configurated e-nose setup was also tested in the field for its effectiveness in detecting VOCs emitted by the fruiting bodies of *F. siamensis*.

This research explored the use of nitrogen gas for the purpose of creating a uniform and consistent baseline to address the possible differences brought by variable ambient air composition in different operation environments, such as the contributions of automobile and industrial waste volatiles during field applications of the e-nose at roadsides or around industrial parks. The testing of selected volatiles using nitrogen for the baseline gas significantly improved the sensitivity and specificity of e-nose detection, as indicated by the distinctive clustering at even lower concentrations. With nitrogen for the baseline gas, the e-nose could clearly differentiate the three selected volatiles from air at 0.05 ppm, and TUR, FAR, and AA were distinctively clustered with an MDist greater than 5.000 for each pairwise comparison (Figure 2D). The same was not true for the ambient air as the baseline gas, where the e-nose had lower sensitivity and specificity and was unable to clearly distinguish TUR from FAR and AA (Figure 2C). Further analysis of sensor responses found that more sensors became responsive when nitrogen was used for the baseline gas, which could have further improved clustering resolution. Furthermore, for the same responsive sensors for both nitrogen as the baseline gas and ambient air as the baseline gas, the magnitude of response was far greater for the nitrogen as the baseline gas. The greater number of responsive sensors and higher magnitude of sensor response are proposed to be the main contributors to the better sensitivity and specificity of the e-nose. This improvement in e-nose performance could be due to the inert properties of nitrogen gas. Using ambient air for the baseline gas, the remaining air in the chamber could react differently with various volatiles, hence affecting sensor responses. Nitrogen effectively displaces oxygen as well as other pro-oxidative gases and purges away the impurities of the sensors during baseline gas recording. For the same reason, nitrogen gas is commonly used for instrument purging in the food [40], aerospace [41], and science [42] industries. A consistent low baseline allows a higher sensitivity to minute changes in volatile profiles, resulting in the e-nose’s ability to better differentiate between sample classes. Pure nitrogen is readily available, and our use of a gas pouch makes it convenient to use nitrogen for field analysis.

The e-nose with nitrogen for the baseline gas could also specifically differentiate fungal mycelia cut plugs of the endemic wood-rot fungus *F. siamensis* from those of another endemic wood-rot fungi, *R. microporus*. Such e-nose differentiation was supported by the distinctive volatile profiles revealed by SPME GC-MS. The specific and high abundance of the *F. siamensis* signature volatiles, such as 1,2,4,5-tetrachloro-3,6-dimethoxybenzene could have underlined the good differentiation by the e-nose. There was a further evaluation of the usage of the e-nose for detecting *F. siamensis* in soil. The same two fungal cultures were mixed with autoclaved soil in different ratios before the e-nose test. It was found that mixing autoclaved soil with cut plugs of fungal culture significantly reduced classification specificity between the *F. siamensis* and *R. microporus* samples. While e-nose could still differentiate the control PDA media cut plug sample from the fungal mycelia cut plug samples; *F. siamensis* and *R. microporus* samples were clustered together. The soil volatiles could have diluted/masked the pathogen-specific volatiles during e-nose testing, resulting in the poor resolution between the two pathogen samples. As such, soil sampling may not be an ideal sampling type for the pathogen-specific field diagnosis of *F. siamensis* with the Cyranose 320.

For field applications with the e-nose, there is a need to find the balance between keeping sample incubation time short for quick analysis and ensuring a sufficient quantity of volatiles above the e-nose detection threshold in the headspace for accurate diagnosis. Our research investigated different incubation time periods and their efficiency in differentiating *F. siamensis* fruiting bodies from reference samples. A 15-min incubation time prior to e-nose analysis was found sufficient enough, with a comparable differentiation result for an incubation time of 5 days (Figure 4C,D). During the field testing, all six *F. siamensis* fruiting bodies of varying sizes collected from two host tree species were successfully identified to a *F. siamensis* sample in Model A (YF156FB1) with a 15-min incubation time. However, the identification of the smaller samples, YF1-B and YF2-B, had lower confidence than the bigger samples, YF1-A and YF2-A, which had the highest possible identification confidence level (*****). The smaller-sized fruiting bodies within the bag may have affected the quantity of volatiles in the headspace, hence resulting in a lower confidence level during the diagnosis test. An exception to this was the small piece of fruiting body R1-B, which gave an equally high identification confidence level (*****) as the larger fruiting body piece R1-A, possibly due to more volatiles released by fruiting bodies grown on rain trees.

Nonetheless, all samples of *F. siamensis* released a significant amount (>10% of total volatiles) of 1,2,4,5-tetrachloro-3,6-dimethoxybenzene. These samples included *F. siamensis* in culture, fruiting bodies, or diseased tissue samples collected from varying tree species. Since this compound is totally absent or present in very minute amounts (<1%) in healthy wood tissues and fruiting bodies of other fungi, this compound is regarded a signature volatile for this fungus species. With a close association with *F. siamensis*, this signature significantly enhanced the e-nose’s differential power in field applications, facilitated the effective identification of *F. siamensis* fruiting body samples. It can also be targeted for *F. siamensis* diagnosis by other methodologies.

In addition to different fungi pathogens, our study also identified several additional factors that may influence the e-nose clustering/differentiation of samples:

Firstly, whilst the use of the Normalisation 1 algorithm tool was intended to mitigate the effect of the volatile concentration on clustering results, this paper showed that volatile quantity changes continued to have an effect on the CDA clustering results (Figure 6). As such, a quantity-dependent variable response further complicates e-nose diagnosis, for which we originally only expected chemical structure-specific classification. On one hand, it suggests the capability of quantitative analysis by the Cyranose 320 e-nose, which has not been reported so far. This feature can possibly be explored to quantify a target volatile. On the other hand, quantity-sensitive clustering could complicate e-nose diagnosis and should be taken into consideration when designing a test. It is suggested that the sample incubation time for headspace volatile concentration equilibration needs to be kept constant to ensure a fair comparison during diagnosis with an existing set of sample databases of training models.

Secondly, *F. siamensis* fruiting body samples from different tree species could be distinctly classified by the e-nose, consistent with the variable volatile profiles revealed by GC-MS analysis (Table 3). Fungal VOCs have been observed to diversify according to their fungal lifestyle or biological activity, such as trophic mode, plant substrate utilization (i.e., root, shoot, leaf litter), and host type (i.e., herbaceous or tree-associated) [20]. Fungal volatile emissions have also been identified to change according to the environmental conditions, such as host plant tissue conditions or with organisms that have established a mutualistic relationship with the fungi [43]. Our result provided one example of variable volatile profiles associated with fungus-host tree interaction.

Thirdly, molecularly identical fungal fruiting bodies with different morphology could also result in varying volatiles being released. *Agaricales* such as *F. siamensis* have been identified to show high levels of morphological plasticity in their fruiting body shape that are influenced by environmental and physiological conditions [44]. SPME GC-MS results show that the volatile profiles of two *F. siamensis* from similar host tree species (CE149FB and CE151FB) were different, with the identification of two more volatiles, 3,5-bis(chloranyl)-2,4-dimethoxy-6-methyl-phenol and 1,4-dichloro-2,5-dimethoxybenzene, that are present in CE149FB, but not in CE151FB. E-nose classification also could distinguish CE149FB from CE151FB (Figure 4C,D). Morphologically, CE149FB and CE151FB vary in color, with CE149FB having a dark brown underside while CE151FB has a lighter orange-brown underside. Hence, morphologically different fruiting bodies of the same species could have different volatile profiles. It is also supported by studies by Kielak et al. [45] and Cellini et al. [21], in that the changes in bacteria composition according to the stage of decay in the plant material could contribute to the differences in volatile emissions, and hence affect the classification. Therefore, diagnosis models may require a comprehensive set of trained sample classes that include various fruiting body morphologies.

For developing e-nose diagnosis, this study primarily utilized samples collected from trees in a late stage of wood decay, with visible symptoms, the presence of fruiting bodies, and the presence of cavities that were detectable by resistance drilling. Whilst the e-nose has been proven to be capable of qualitatively differentiating *F. siamensis* culture, fruiting bodies, and decayed tissues from healthy wood, *R. microporus*, and *G. australe* samples, more work is required for the early-stage detection of *F. siamensis* in trees, and in the absence of a fruiting body. There is a need to build a more comprehensive model to include samples at different infection stages from the multiple host tree species. Understanding stage-specific volatile profiles for *F. siamensis* infection of each host tree will also be critical to diagnose incipient (early-stage) fungal infection, either by e-nose or other volatile-based detection such as SPME GC-MS.

In summary, this paper demonstrates that the Cyranose 320 e-nose with nitrogen for the baseline gas could effectively differentiate *F. siamensis* culture, fruiting bodies, and decayed wood from reference samples. The paper also showcases the good potential for the e-nose in non-invasive diagnosis due to the minimal operation cost per sample (20 mL of nitrogen, sample bag), its reusability due to its purging system to clean its sensors prior to each sampling, and its quick sample preparation (15 min) and fast diagnosis response time per reading (less than 1 min). Using nitrogen for the baseline gas significantly improves sensor responses. The same setup was successfully used in the field to identify *F. siamensis* fruiting bodies collected from two host tree species. The e-nose differentiation was consistent with the volatile profiles characterized by the SPME GC-MS. The presence of the signature volatile for *F. siamensis* very likely underlines such species-specific diagnosis. While the e-nose has the potential to quickly and inexpensively conduct diagnosis of wood-rot fungus *F. siamensis*, there are various limitations to the current technology such as: (1) the e-nose having a limited sample capacity (six) for model building for usage during the ‘Identify’ function; (2) volatile concentrations may complicate the e-nose classification and diagnosis confidence; (3) fungal pathogen-host tree interactions that result in tree-species-specific volatile profiles may create complications in developing and optimizing the e-nose model for pathogen-specific diagnosis.

Our findings add more understanding to the potential of volatile-based detection techniques for assessing urban tree health and contribute towards the advancement of urban forestry management strategies.

## Figures and Tables

**Figure 1 sensors-23-04538-f001:**
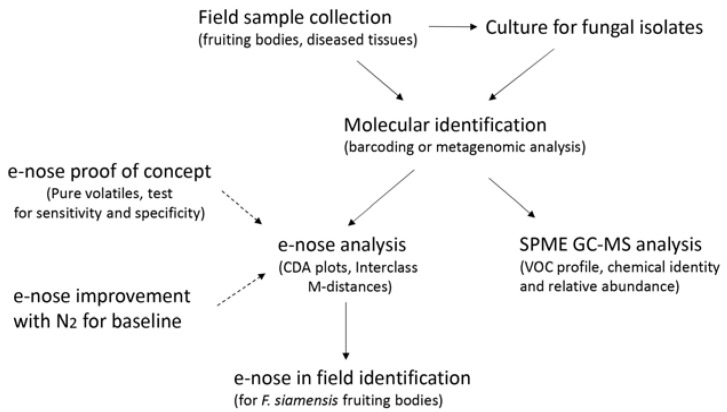
Experiment flow for this study.

**Figure 2 sensors-23-04538-f002:**
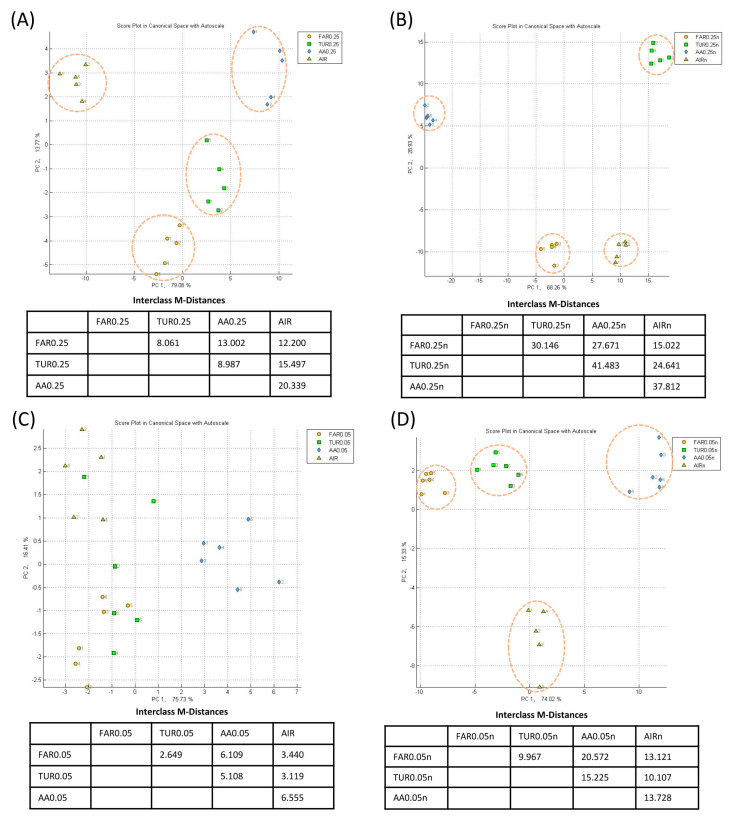
Nitrogen for the baseline improves e-nose sensitivity and specificity. CDA plots with Interclass M-Distance readings for 0.25 ppm of Farnesene (FAR0.25), Turpentine (TUR0.25), Acetic Acid (AA0.25), and air from biosafety cabinet (AIR) volatile samples using (**A**) ambient air and (**B**) nitrogen as the baseline gas (samples have additional suffix –n); and for 0.05 ppm VOC samples (FAR0.05, TUR0.05, AA0.05) using (**C**) ambient air and (**D**) nitrogen as the baseline gas. Samples inside a dashed-line ellipse indicate that the Interclass M-Distance is greater than 5.000 and thus form a distinctive cluster.

**Figure 3 sensors-23-04538-f003:**
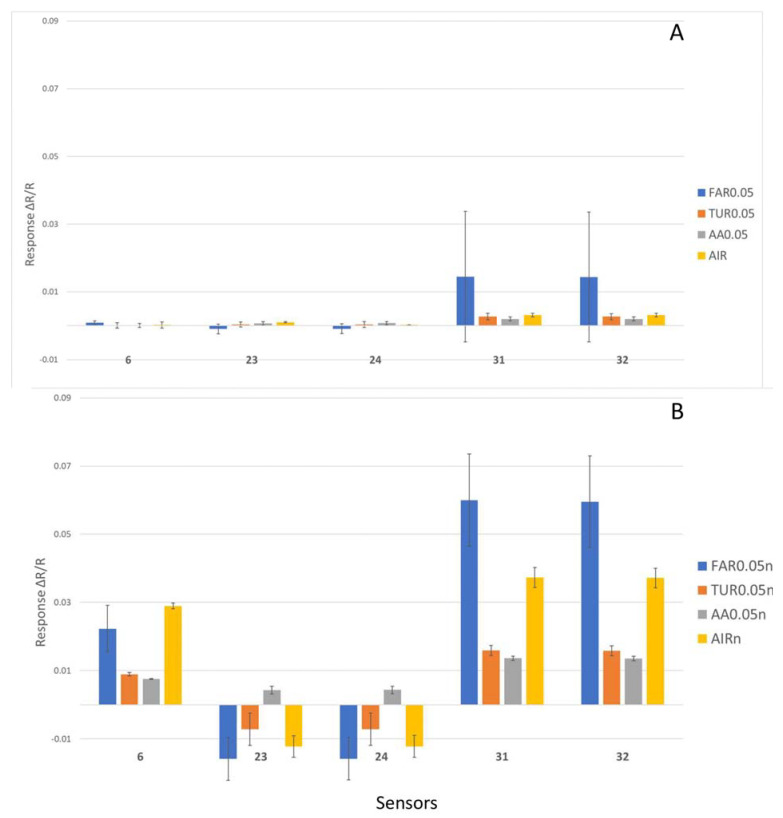
Better sensor responses using nitrogen as the baseline gas. Mean response ΔR/R_o_ of the selected active sensors 6, 23, 24, 31, and 32 for volatile chemicals Farnesene (FAR0.05), Turpentine (TUR0.05), Acetic Acid (AA0.05) at 0.05 ppm and air from biosafety cabinet (AIR) using (**A**) ambient air and (**B**) nitrogen as the baseline gas (samples have additional suffix -n).

**Figure 4 sensors-23-04538-f004:**
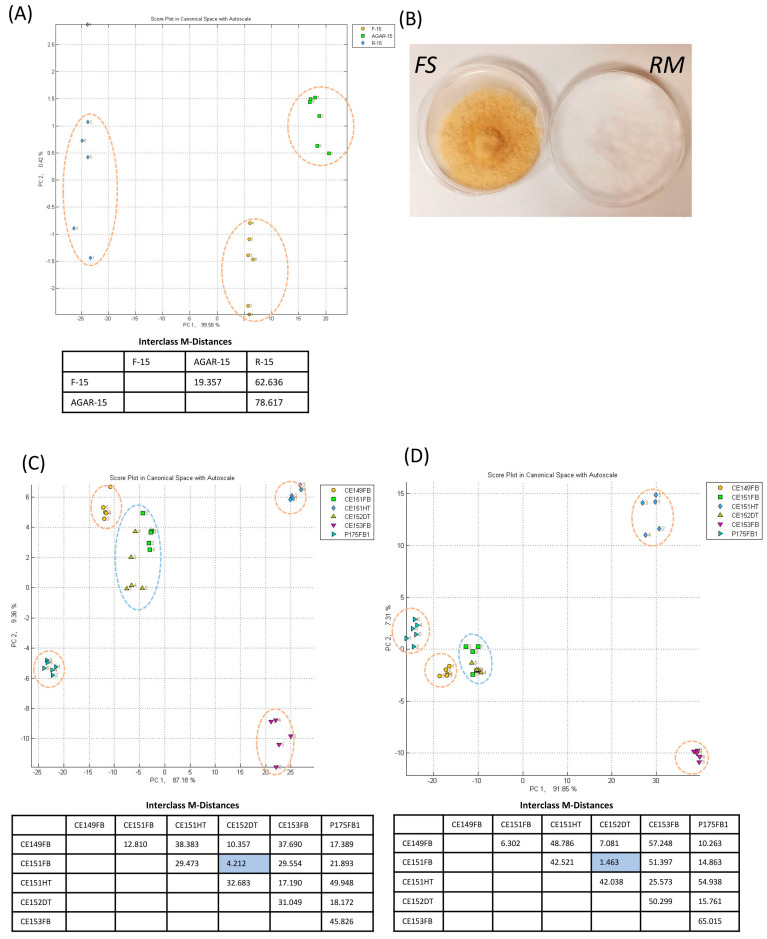
E-nose specifically detects *F. siamensis* samples. (**A**): CDA plot of e-nose clustering with nitrogen as the baseline gas for fifteen *F. siamensis* pure fungal mycelia cut plug samples (F-15) against that of *R. microporus* (R-15); (**B**): cultures of *F. siamensis* (FS) and *R. microporus* (RM); (**C**): CDA plot of *F. siamensis* fruiting bodies (CE149FB, CE151FB) and diseased tissue (CE152DT) against *G. australe* fruiting body (CE153FB), *R. microporus* fruiting body (P175FB1) with 15 min of incubation; (**D**): CDA plot for the same samples after a 5-day incubation period. An orange-dashed line ellipse indicates a distinctive cluster for a single sample, with Interclass M-Distance greater than 5.000 from any other sample. A blue dashed-line ellipse indicates a cluster of two samples with Interclass M-Distance less than 5.000, which is also highlighted in the table.

**Figure 5 sensors-23-04538-f005:**
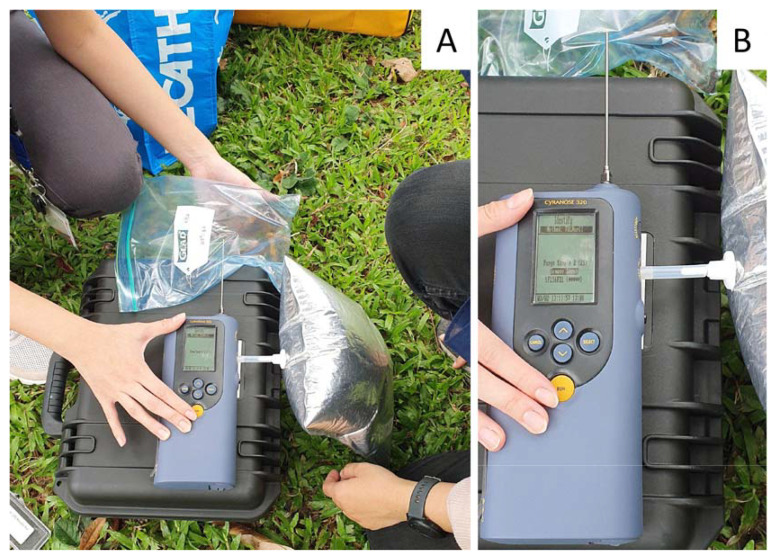
Real-time field testing with the e-nose. (**A**) The e-nose setup with the nitrogen pouch; (**B**) In-field identification of a *F. siamensis* fruiting body with high confidence after 15-min incubation time.

**Figure 6 sensors-23-04538-f006:**
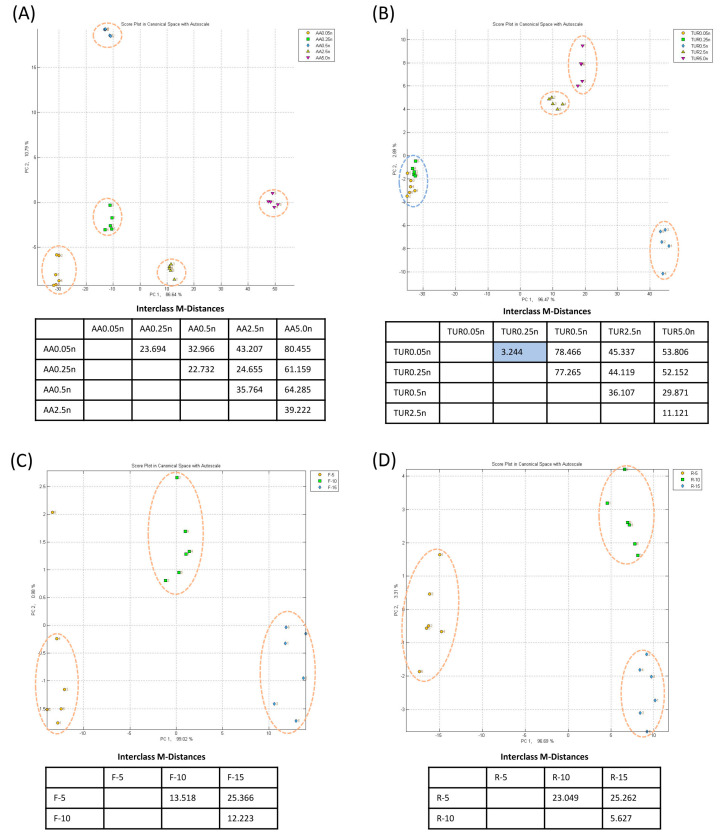
Volatile quantity-based clustering by the e-nose. CDA plots with Interclass M-distances for different concentrations of AA (**A**) and TUR (**B**), and for different numbers (five, ten, and fifteen) of fungal mycelia cut plugs for *F. siamensis* (**C**) and *R. microporus* (**D**). An orange-dashed line ellipse indicates a distinctive cluster for a single sample, with Interclass M-Distance greater than 5.000 from any other sample. A blue dashed-line ellipse indicates a cluster of two samples with Interclass M-Distance less than 5.000, which is also highlighted in the table.

**Table 1 sensors-23-04538-t001:** List of samples for e-nose and SPME GC-MS analysis and their molecular identities.

Date of Collection	Type of Sample	Host Tree	Sample Marking	Location and Remarks	Molecular Identity/GenBank Accession No.
11 October 2021	fruiting body	*Syzygium grande*	SG145FB	1°18′38.4″ N 103°49′58.6″ E Tree has multiple fruiting bodies at ground level, the largest fruiting body was collected	*F. siamensis*/OQ558845
16 December 2021	fruiting body	*Casuarina equisetifolia*	CE149FB	1°17′20.3″ N 103°46′12.6″ E	*F. siamensis*/OQ558844
23 December 2021	diseased tissue	*Casuarina equisetifolia*	CE149DT2	1°17′20.3″ N 103°46′12.6″ E Decayed wood from cut open stump	^ 79.4% *F. siamensis*/OQ572588^ 3.8% *G. australe*/OQ572592
23 December 2021	diseased tissue	*Casuarina equisetifolia*	CE149DT5	1°17′20.3″ N 103°46′12.6″ E Decayed wood behind a *F. siamensis* fruiting body	^ 90.5% *F. siamensis*/OQ572588
16 December 2021	healthy tissue	*Casuarina equisetifolia*	CE151HT	1°17′20.3″ N 103°46′12.6″ E Wood obtained from healthy-looking area of the root	-
16 December 2021	fruiting body	*Casuarina equisetifolia*	CE151FB	1°17′20.3″ N 103°46′12.6″ E	*F. siamensis*/OQ558844
16 December 2021	diseased tissue	*Casuarina equisetifolia*	CE152DT	1°17′20.3″ N 103°46′12.6″ E Decayed wood behind a *F. siamensis* fruiting body	*^ 96.0% F. siamensis*/OQ572588^ *0.4% G. australe*/OQ572592, OQ572594
16 December 2021	fruiting body	*Casuarina equisetifolia*	CE153FB	1°17′20.3″ N 103°46′12.6″ E	*G. australe*/OQ572592, OQ572594
25 February 2022	fruiting body	*Peltophorum pterocarpum*	YF156FB1	1°19′14.8″ N 103°49′07.8″ E	*F. siamensis*/OQ558847
25 February 2022	fruiting body	*Peltophorum pterocarpum*	YF157FB	1°19′14.8″ N 103°49′07.8″ E	*F. siamensis*/OQ558848
25 February 2022	fruiting body	*Samanea saman*	R159FB	1°16′58.5″ N 103°49′53.9″ E	*F. siamensis*/OQ558848
25 February 2022	fruiting body	*Peltophorum pterocarpum*	YF160FB	1°16′56.7″ N 103°49′52.4″ E	*F. siamensis*/OQ558848
17 May 2022	fruiting body	*Sabal palmetto*	P175FB1	1°19′02.9″ N 103°46′09.2″ E	*R. microporus*/OQ558868
16 September 2022	fruiting body	*Sabal palmetto*	P175FB3	1°19′02.9″ N 103°46′09.2″ E	*R. microporus*/OQ558868
16 September 2022	diseased tissue	*Tabebuia rosea*	TR190DT2	1°19′18.4″ N 103°55′31.8″ E Decayed wood behind *F. siamensis* fruiting body	*^ 95.8% F. siamensis*/OQ572588^ *3.60% Fomitiporia bannaensis*/OQ572582

^ relative abundance of wood decay fungi in the diseased tissue samples identified through ITS1 metagenomics analysis.

**Table 2 sensors-23-04538-t002:** E-nose sampling method settings.

	Time (s)	Pump Speed
Baseline purge	10	Medium (120 mL/min)
Sample draw	10	Medium (120 mL/min)
Air intake purge	5	High (180 mL/min)
Sample gas purge	30	High (180 mL/min)

**Table 3 sensors-23-04538-t003:** Volatiles identified by SPME GC-MS. CAS # refers to the CAS Registry Number assigned by the Chemical Abstracts Service, USA. Relative abundance of each volatile in the samples is represented by: * for 0.1–1%; ** for 1–5%; *** for 5–10%; **** for 10–20%, and ***** for >20%. Abbreviations: *FS* for *Fulvifomes siamensis*; *RM* for *Rigidoporus microporus*; *GA* for *Ganoderma australe*; FB for fruiting body; DT for diseased tissue; CE for *Casuarina equisetifolia*; SG for *Syzygium grade*; YF for *Peltophorum pterocarpum*; R for *Samanea saman*; P for *Sabal palmetto*; ^ relative abundance of wood decay fungi found in the diseased tissue samples identified through ITS1 metagenomics analysis.

	Samples	CAS #	FS Culture	RM Culture	SG 145FB (FS)	YF157FB (FS)	R159FB (FS)	CE149FB (FS)	CE151FB (FS)	CE153 FB (GA)	P175 FB3 (RM)	CE149DT2 (79% FS, 3% GA) ^	CE149DT5 (90% FS) ^	TR190DT2 (95% FS) ^
Volatiles														
2,4-bis(chloranyl)-1,5-dimethoxy-3-methyl-benzene		997271-90-6										**	***	
3,5-bis(chloranyl)-2,4-dimethoxy-6-methyl-phenol		997329-33-1				**	*	**				***	**	**
Acetic acid, 2-phenylethyl ester		000103-45-7	**											
Benzene, 1,2,4,5-tetrachloro-3,6-dimethoxy-		000944-78-5	*****		*****	*****	*****	*****	*****			****	**	*****
Benzene, 1,2,4-trichloro-5-nitro-		000089-69-0				*****								
Benzene, 1,4-dichloro-2,5-dimethoxy-		002675-77-6				**	***	*				****	*****	**
beta-Bisabolene		000495-61-4	*		*									**
(E)-.beta.-Farnesene		018794-84-8	*****			*	*							***
2,7-dichloro-1-methoxydibenzofuran		067061-60-3	**											
Methyl 2,6-Dichloro-4-methoxybenzoate		094278-65-6	**											
Benzoic acid, methyl ester		000093-58-3		***										
Cetrimonium Bromide		000057-09-0		*****										
Disulfide, dimethyl		000624-92-0									****			
Hexadecane, 1-chloro		004860-03-1		**										
Phenylethyl Alcohol		000060-12-8									**			
Tributylamine		000102-82-9		***										
Butyrolactone		000096-48-0					**							
(3aS,8aS)-6,8a-Dimethyl-3-(propan-2-ylidene)-1,2,3,3a,4,5,8,8a-octahydroazulene		395070-76-5				*	*					*		**
2,2,4-Trimethyl-1,3-pentanediol diisobutyrate		006846-50-0		**										
3-(4-Nitro-phenylsulfanyl)-propionic acid		997299-58-6										****		
3-Methyl -6-(3-methylthiophen-2-yl)-[1,2,4]triazolo[3,4-b][1,3,4]thiadiazole		997329-32-4										**		
3-Octanone		000106-68-3	*	*			**		*			*		
Acetic acid		000064-19-7			***					**				
Benzene, 1-(1,5-dimethyl-4-hexenyl)-4-methyl		000644-30-4	*	*		*								
Benzene, 1,3-dimethyl-		000108-38-3					*				**			*
Decanal		000112-31-2												
Diethyl Phthalate		000084-66-2			*					****				
Docosane		000629-97-0								**				
Nonanal		000124-19-6			*		*						*	
Phenol, 2,3,5,6-tetrachloro-4-methoxy		000484-67-3			*		**					*	*	
Toluene		000108-88-3		*	*		*	*		*	**			*

**Table 4 sensors-23-04538-t004:** E-nose real-time field diagnosis of *F. siamensis* fruiting bodies.

Sample ID	Collection Date	Host Tree	Fruiting Body Weight (g)	E-Nose Diagnosis Result (Confidence)	Molecular Identity by Barcoding
R1-A	2 March 2023	*Samanea saman*	104.50	YF156FB1 (*****)	*F. siamensis*
R1-B	2 March 2023	*Samanea saman*	5.33	YF156FB1 (*****)	*F. siamensis*
YF1-A	2 March 2023	*Peltophorum pterocarpum*	336.14	YF156FB1 (*****)	*F. siamensis*
YF1-B	2 March 2023	*Peltophorum pterocarpum*	12.14	YF156FB1 (*)	*F. siamensis*
YF2-A	2 March 2023	*Peltophorum pterocarpum*	191.17	YF156FB1 (*****)	*F. siamensis*
YF2-B	2 March 2023	*Peltophorum pterocarpum*	15.44	YF156FB1 (***)	*F. siamensis*
YF2 Tree Stump	2 March 2023	*Peltophorum pterocarpum*	Negative Control	Unknown	-

Confidence of results: lowest confidence (*), medium confidence (***), highest confidence (*****).

## Data Availability

More data supporting reported results can be available upon request.

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
