# Peer review of "Volatile-Based Diagnosis for Pathogenic Wood-Rot Fungus Fulvifomes siamensis by Electronic Nose (E-Nose) and Solid-Phase Microextraction/Gas Chromatography/Mass Spectrometry"

_sensors, 2023, doi:10.3390/s23094538_

Round 1
Reviewer 1 Report
Dear Jhing Yein Tan, Ziteng Zhang, Hazirah Junin Izzah, Yok King Fong, Daryl Lee, Marek Mutwil, and Yan Hong,
I have reviewed your manuscript, titled "Volatile-based Diagnosis for Pathogenic Wood-rot fungus Fulvifomes siamensis by Electronic Nose (e-nose) and Solid-Phase Microextraction/Gas Chromatography/Mass Spectrometry," which was submitted to the Journal Sensors (ISSN 1424-8220) with the manuscript ID sensors-2364044. I must say that the research work presented in your article is impressive and provides valuable insights into the application of electronic nose technology for the diagnosis of the pathogenic wood-rot fungus Fulvifomes siamensis.
However, I have noticed that the figures and tables in your manuscript could be rearranged again to improve the clarity of the presentation. I suggest that you take some time to revise these sections plus some comments below to ensure that they effectively communicate the results and findings of your research
Overall, your manuscript makes a significant contribution to the field of Advances in Intelligent Biosignals Processing and Analysis, and I am happy to recommend it for publication in the Journal Sensors. Thank you for submitting your work to our journal, and I look forward to seeing more research work from you in the future.
Best regards
Comments:
- The PC2 figures are too large. Please make them more organized.
- The Interclass M-Distances tables have an extra empty row and column.
- The number of digits after "MDist" should be consistent throughout the paper.
- Including a diagram to explain the experimental flow in Figure 1 would enhance the clarity of the paper.
- The results of the concentrations between 0.05 and 5 ppm should be included in Figure 1 as proof of concept, as these are the concentrations used to test the sensitivity of the method.
- The paper needs more explanations to clarify the performance factors of the e-noise, including response time and reusability.
- The evaluation system used to diagnose the infection as small, medium, or high confidence measurement units should be supported with strong references to determine if it is a universal evaluation system.
Author Response
Dear reviewer, we thank you for your appreciation and the many helpful comments and suggestions. We have made changes based on the comments and suggestions from the reviewers. The attached file has the point-to-point responses to your comments.

Reviewer 2 Report
This manuscript investigated the volatile-based diagnosis for pathogenic wood-rot fungus by commercial e-nose and SPME GC-MS. The results found that nitrogen purging can help the improvement of baseline and sensor responses of e-nose. And the e-nose has good specificity towards different tree species, which is also validated and supported by SPME GC-MS analysis.
A few issues should be addressed or discussed before consideration of publication.
(1) This method is based on the characteristic VOCs for discriminating the diseased trees, and the background VOCs of the tree itself may contribute much more than the fungus interaction. Is there some more supporting information about the VOCs of healthy trees and diseased trees?Is there characteristic VOCs resulting from the fungus contamination? Do we need to determine the identity of those VOCs? After all, here we can only see that different groups can be discriminated in the PCA plots.
(2) For the VOCs combination, if the concentration of ingradient VOCs varies, how will it affect the responses of e-nose?
(3)It would be better to summarize the advantage of e-nose for the tree checking in the part of disccussion, such as the cost, the time, the sensitivity, and specificity.
Author Response

(The authors gave the same response as above.)
